# Hydrothermal Synthesis of Nitrogen, Boron Co-Doped Graphene with Enhanced Electro-Catalytic Activity for Cymoxanil Detection

**DOI:** 10.3390/s21196630

**Published:** 2021-10-05

**Authors:** Codruța Varodi, Florina Pogăcean, Maria Coros, Lidia Magerusan, Raluca-Ioana Stefan-van Staden, Stela Pruneanu

**Affiliations:** 1National Institute for Research and Development of Isotopic and Molecular Technologies, Donat Street No. 67-103, 400293 Cluj-Napoca, Romania; codruta.varodi@itim-cj.ro (C.V.); florina.pogacean@itim-cj.ro (F.P.); lidia.magerusan@itim-cj.ro (L.M.); 2Laboratory of Electrochemistry and PATLAB, National Institute of Research for Electrochemistry and Condensed Matter, 060021 Bucharest, Romania; ralucavanstaden@gmail.com; 3Faculty of Applied Chemistry and Material Science, University Politehnica of Bucharest, 060021 Bucharest, Romania

**Keywords:** nitrogen, boron co-doped graphene, cymoxanil, electrochemical detection

## Abstract

A sample of nitrogen and boron co-doped graphene (NB-Gr) was obtained by the hydrothermal method using urea and boric acid as doping sources. According to XRD analysis, the NB-Gr sample was formed by five-layer graphene. In addition, the XPS analysis confirmed the nitrogen and boron co-doping of the graphene sample. After synthesis, the investigation of the electro-catalytic properties of the bare (GC) and graphene-modified electrode (NB-Gr/GC) towards cymoxanil detection (CYM) was performed. Significant differences between the two electrodes were noticed. In the first case (GC) the peak current modulus was small (1.12 × 10^−5^ A) and appeared in the region of negative potentials (−0.9 V). In contrast, when NB-Gr was present on top of the GC electrode it promoted the transfer of electrons, leading to a large peak current increase (1.65 × 10^−5^ A) and a positive shift of the peak potential (−0.75 V). The NB-Gr/GC electrode was also tested for its ability to detect cymoxanil from a commercial fungicide (CURZATE MANOX) by the standard addition method, giving a recovery of 99%.

## 1. Introduction

Pesticides are chemical compounds used to extirpate pests, including insects, rodents, fungi and weeds. Over 1000 different pesticides are used around the world [1]. Pesticides include active and inert ingredients which may be carcinogens or toxic. Prevention of the negative effects of pesticides requires precise control of their remaining content in agricultural products, food, soil, and water [2]. Reliable analytical procedures are therefore needed for their correct determination. Electrochemical methods are recognized for their superior sensitivity, lower detection limits, and cost-efficiency. Moreover, their facile operation, rapid analytical response, absence of sample pre-treatment, and miniaturization, makes them extremely suitable for on-site analysis. Recently, great efforts have been devoted to the quest of finding appropriate materials for electrochemical detection of pesticides, and other emerging pollutants [3]. Electrode modification serves as an efficient alternative for decreasing the over-potential of electrochemical reactions and pre-concentrating capacity for some analytes. In addition, it determines the generation of an electrode-solution interface which significantly improves the signal. Carbon materials are irreplaceable in electrochemical sensing for many reasons: they are good electronic conductors, cheap, abundant, easy to work with, chemically inert and suitable for making composites. In particular, graphene-based nanomaterials have become intensively used in environmental analysis as electrochemical sensors or biosensors [4]. Various pesticides have been detected using graphene-modified electrodes, such as: carbaryl (an insecticide) and paraquat (one of the most widely used herbicides)- with a graphene-modified boron-doped diamond electrode [5]; profenofos, fenitrothion and parathion methyl (organophosphate insecticides) with graphene-Mn electrodes [6] and a screen-printed silver electrode modified with a graphene nanoplatelet and a zirconia coating, respectively [7]; or 2, 4, 6 trichlorophenol (fungicide, herbicide, insecticide, antiseptic) with a glassy carbon electrode modified with molybdenum bismuth vanadate impregnated on graphene oxide [8]. Some interesting review papers on graphene-based materials used for pesticides detection are also available in the literature [9,10,11]. The use of graphene-based nanomaterials in sensor development for pesticides detection has been recently employed, due to several advantages: large surface area, excellent conductivity and analytical performances [12,13].

Graphene is a special sp^2^ bonded carbon allotrope, which has drawn great attention in the scientific world during the last decade. This is coming from its unique application potential in the fields of nanoelectronics [14], capacitors [15] and sensors [16,17]. Nonetheless, pure graphene with no band-gap displayed limited electro-catalytic activity due to the deficient number of active sites. The heteroatom doping can be used to open the band-gap and tune the Fermi level of graphene [18,19]. The doped carbonaceous materials are especially interesting as they are known to have better electro-catalytic activity in comparison to their un-doped equivalents [20,21]. Recent works reported that dual-doped graphene with two types of heteroatoms (e.g., N, B [22,23], N, S [24,25,26], or N, P [27,28]) presented improved electro-catalytic activity due to their synergistic effect compared to single atom-doped graphene. Nitrogen, having a larger electronegativity than carbon (N: 3.04, C: 2.55, respectively,) could induce a charge redistribution of carbon thus generating an improved electronic conductivity and formation of supplementary active site [29]. On the other side, boron has a smaller electronegativity than carbon (2.04). It was suggested that boron doping improves electrochemical performance of graphene mainly because of the B-C structure which serve as active site [30]. Based on the benefits induced by N and B doping of graphene, there are various synthesis methods developed to prepare these type of materials [31]. Among them, hydrothermal synthesis has the advantage of being facile, efficient and feasible for the simultaneous N and B co-doping of graphene.

Cymoxanil (2-cyano-N-[(ethylamino) carbonyl]-2-(methoxyimino) acetamide) is an aliphatic nitrogen fungicide. Cymoxanil (CYM) is employed both as a curative and preventive foliar fungicide. In Europe, it is used on potato, tomato, hop, sugar beet and grape crops [32]. The most frequently used methods for determination of cymoxanil in water, fruits or vegetables are: high-performance liquid chromatography with mass spectrometry or UV detection [33,34], and gas chromatography with a nitrogen-phosphorus detector [35]. An electrochemical method for cymoxanil determination was also described, using square-wave stripping voltammetry at the mercury electrode [36]. To the best of our knowledge, no paper has been published about cymoxanil determination using an electrode modified with nitrogen and boron co-doped graphene.

The aim of this work is to present a method for cymoxanil determination by linear sweep voltammetry (LSV) and amperometry using a glassy carbon electrode modified with nitrogen, boron co-doped graphene, synthesized by the hydrothermal method. The performance of the electrode is compared with that of the bare glassy carbon electrode (GC) and the applicability of the method is verified by detecting CYM in a commercial product by the standard addition method.

## 2. Materials and Methods

### 2.1. Chemicals and Materials

All reagents were of analytical grade and used without further purification. Graphene oxide (GO) was synthesized using a modified Hummers method as previously reported [37]. Urea was purchased from Alfa Aesar (Kandel, Germany) and boric acid from Adra Chim (Bucharest, Romania). CURZATE MANOX was purchased from Cluj-Napoca, Romania and was produced by Chemical Independent Group (Câmpia Turzii, Romania).

### 2.2. Instruments

The sample was morphologically and structurally characterized using scanning electron microscopy (SEM-Hitachi SU 8230, Tokyo, Japan), X-ray powder diffraction (Bruker D8 Advance Diffractometer, Karlsruhe, Germany), Raman spectroscopy (NTEGRA Spectra platform, placed on a NEWPORT RS4000 optical table and equipped with a SOLAR TII confocal Raman spectrometer coupled with an Olympus IX71 microscope in two different configurations) (Moscow, Russia), and X-ray photoelectron spectroscopy (XPS-SPECS spectrometer equipped with a dual anode X-ray source AlMg, a PHOIBOS 150 2DCCD hemispherical energy analyzed and a multi Channeltron detector) (Berlin, Germany).

Electrochemical measurements were performed with an AUTOLAB-302N Potentiostat/Galvanostat (Utrecht, The Netherlands) coupled with a computer and a three-electrode cell. The experimental data were interpreted with NOVA 1.11 software. Ag/AgCl (3 M KCl) was employed as a reference electrode and platinum foil (1 cm^2^ area) as a counter-electrode. The working electrode was either bare glassy carbon (GC) or GC modified with the synthesized graphene sample (NB-Gr/GC). The linear sweep voltammetry technique was employed for studying the redox activity of cymoxanil (scan rate 10 mV/s; potential range: −1.5… +0.6 V vs Ag/AgCl). The amperometric measurements were recorded at a potential of −0.8 V vs. Ag/AgCl, in pH 6 Britton–Robinson buffer. A WTW-Multi 3320 pH meter (Weilheim, Germany) was used for pH measurements.

### 2.3. Hydrothermal Synthesis of NB-Gr Sample

GO (700 mg) was dispersed in 120 mL distilled H_2_O by sonication (1 h). After that, 1000 mg urea and 1000 mg H_3_BO_3_ were added and the suspension was stirred for 1 h at room temperature. The obtained mixture was then poured into a 250 mL autoclave and placed in the oven at 180 °C for 12 h. After cooling to room temperature, the sample was filtered, washed with distilled water and dried by lyophilization. An amount of 320 mg of the final product resulted after lyophilization. The nitrogen and boron co-doped graphene sample was then denoted NB-Gr.

## 3. Results

The morphological aspects of the synthesized sample were investigated by SEM technique. The SEM micrographs with different magnifications showed thin graphene sheets with wrinkled and folded areas (Figure 1a,b). The wave-like morphology of the sample may be due to the heteroatoms or defects present in the graphene plane. 

The structure of the NB-Gr sample was studied by X-ray powder diffraction (Figure 2). The XRD pattern of the sample shows a diffraction peak at 20~25°, corresponding to an interlayer spacing of about 0.358 nm. The size of graphene crystallites (D) was calculated using the Scherrer equation and was found to be 1.783 nm. The average number of layers was five.

The structural disorder degree in the co-doped graphene sample was determined using Raman spectroscopy (Figure 2b). All the characteristic graphene bands are visible in the Raman spectrum of the sample: the defect (D) band at ~1356 cm^−1^; the graphite band (G) at ~1608 cm^−1^ and the 2D band at ~2690 cm^−1^. D band intensity is higher than the relative intensity of the G band indicating that B and N atoms were introduced into the lattice of graphene creating more defects [38]. The I_D_/I_G_ ratio gives an indication of the defect-free domains and is related to the in-plane crystallite size (L_a_) as shown by Equation (1) [39]: (1)$$L_{a}\left(\mathrm{nm}\right)=\frac{560}{E_{l}^{4}}{\left(\frac{I_{D}}{I_{G}}\right)}^{-1}$$
where E_l_ represents the laser excitation energy (2.33 eV). 

The relative high intensity ratio (I_D_/I_G_ = 1.066) of the NB-Gr sample may be attributed to more defects generated by nitrogen and boron co-doping, resulting in highly disordered graphene nanosheets. These defects can provide considerable active sites for electrochemical detection, as shown in the next section.

To investigate the doping of N and B in the graphene, the XPS analysis was carried out (Figure 3). 

The core level high-resolution C 1s spectrum (Figure 3a), was deconvoluted into five different components accordingly: sp^2^ and sp^3^ carbon frameworks (284.2 eV and 285.1 eV, respectively); C-O/C-N (285.9 eV); C=O (287.1 eV); COOH/O-C=O/C-O-B (288.9 eV) and a small contribution assigned to the π→π* shake-up satellite band of graphitic carbons (290.6 eV). The oxygen atoms are bonded with the unsaturated carbon atoms present at graphene edge sites in the form of C–O (531.1 eV) and C=O/O–C=O (532.9 eV) groups and the high oxygen content confirms the presence of structural defects inside the sample (Figure 3b). The deconvoluted N 1*s* spectrum indicates three distinct components (Figure 3c) with binding energy (BE) at 398.2, 399.7 and 402 eV for the pyridinic (24.9%), pyrrolic (57.4%) and graphitic (17.7%) nitrogen, respectively. In addition, a weak signal from the boron atoms could be discerned from the XPS spectrum (≈192 eV), indicating the successful incorporation of boron into the graphene framework (Figure 3d). Previous theoretical calculations indicate that due to the large electro-negativity difference between boron and oxygen, O_2_ can be easily absorbed by the boron dopant, resulting in the formation of O^2−^, O_2_^−^, and O_2_^2−^ [40]. Furthermore, boron can substitute the carbon atom at the trigonal sites (BC_3_). Accordingly, the high resolution B 1s spectrum was fitted with two components corresponding to BC_2_O (191.1 eV) and BC_3_ (193.4 eV). The low boron doping level of graphene material can be explained due to the electron accepting nature of boron atoms which makes them more affinitive to oxygen compared to carbon [41]. The C/O, C/N and C/B atomic ratio were found to be: 0.79, 4.02 and 132.64. 

## 4. Electrochemical Studies

### The pH Effect on Cymoxanil Detection

The pH effect on the electrochemical signal of cymoxanil was investigated by LSV with NB-Gr/GC electrode, after scanning the potential from +0.6 to −1.5 V (10 mV/s scanning rate), Figure 4a,b. As can be seen in Figure 4a, cymoxanil has two electro-active functional groups, the reducible ketone and nitrile group. The redox process is irreversible, no oxidation peak being observed in the anodic scan. In acidic and neutral solutions (pH 2–pH 7) a well-defined peak appears during the reduction process, while in basic solutions (pH 8–pH 12) the same peak strongly decreased, especially when pH is higher than pK_a_ (pK_a_ = 9.7 ± 0.2) [42]. 

The variation of the modulus of peak current |I_p_| and peak potential |E_p_| with the solution pH can be seen in Figure 5a,b, respectively. In the first case, the strong decrease in peak intensity in alkaline solutions (pH 8–pH 12) indicates that the protonated species of cymoxanil are involved in the redox process. In addition, the peak potential shifts towards more negative values (pH 2–pH 7), which also confirms the H^+^ involvement in the reduction process. For pH 2–pH 7 range, the linear regression equation was determined to be: y = 0.408 + 0.059 × pH. Above pH 8, a plateau was reached which correlates very well with the pK_a_ of cymoxanil molecules (pK_a_ = 9.7 ± 0.2). Based on such information the optimum pH was selected to be pH 6. Another important reason for selecting a slightly acidic solution for further experiments was related to the hydrolysis of cymoxanil; the molecules hydrolyze in neutral and alkaline aqueous solutions, according to data sheet specifications. In pH 7 solution, the hydrolysis constant is 34 h while in pH 9 solution, the constant is 31 min [42].

Next, the investigation of the electro-catalytic properties of bare GC and NB-Gr/GC electrode was performed (Figure 6). As can be seen in this figure, there are significant differences between the two electrodes, both in terms of peak current and peak potential. In the first case (bare GC) the peak current (modulus) is small, 1.12 × 10^−5^ A, and appears in the region of negative potentials, around −0.9 V. In contrast, when NB-Gr is present on top of the GC electrode it highly promotes the transfer of electrons, leading to a peak current increase up to 1.65 × 10^−5^ A and a positive shift of the peak potential, to −0.75 V. Such changes clearly indicate the excellent electro-catalytic properties of the NB-Gr/GC electrode.

The dependence of cymoxanil peak current on the solution concentration (4 × 10^−5^–10^−3^ M) was measured under the optimized conditions (pH 6 BR) and is presented in Figure 7a,b. As revealed by this figure, the peak current constantly increased with CYM concentration, and the corresponding calibration plot has the linear regression equation expressed by: |I_p_| = 6.94 × 10^−9^ + 0.0134 × C_CYM_ (R^2^ = 0.998). The limit of quantification (LOQ) was determined to be 4 × 10^−5^ M, while the limit of detection (LD) was 1.21 × 10^−5^ M. The modified electrode was also tested for successive measurements and it proved to have a good reproducibility. Between the measurements the electrode was immersed in a mixture of methanol: distilled water for 30 min, then cycled in pH 6 BR until no signal from CYM was observed (about 20 cycles; scan rate 50 mV/s).

Next, the amperometric technique was employed to further test the CYM detection with the NB-Gr modified electrode. Figure 8a shows the amperometric curve recorded in pH 6 BR (potential −0.8 V vs. Ag/AgCl) after the addition of known concentrations of cymoxanil. The corresponding calibration plot (background subtracted) has the linear regression equation ΔI = 2.28 × 10^−6^ + 0.0164 × C_CYM_ and is presented in Figure 8b. By amperometric measurement, both the limit of quantification and limit of detection were considerably smaller. Hence, LOQ was determined to be 5 × 10^−6^ M, while LD was 1.51 × 10^−6^ M.

The NB-Gr/GC electrode was also tested for its ability to detect CYM from a commercial product (CURZATE MANOX; produced by Chemical Independent Group, Romania) by the standard addition method. The product contains 5% CYM, 18% mancozeb (C_4_H_6_N_2_S_4_Mn · C_4_H_6_N_2_S_4_Zn) and 25% Cu_2_(OH)_3_Cl. To 5 mL of pH 6 BR containing a certain concentration of CYM (from CURZATE MANOX) were added known concentrations of CYM from a stock solution (10^−3^ M) (Figure 9a). The generated signals were recorded for each concentration by LSV. After reading the peak current and plotting the calibration curve (Figure 9b) the concentration of CYM was found to be 6.24 × 10^−5^ M, giving a recovery of 99%.

The performances of the NB-Gr/GC electrode in terms of linear range and limit of detection/quantification are compared with those of other types of electrodes, e.g., HMDE- hanging mercury drop electrode; CFRE-composite fiber rod electrode; GCE- glassy carbon electrode (Table 1). According to data sheet specifications of CURZATE MANOX, the optimum concentration used for the treatment of various vegetables and fruits (potatoes, cucumbers, grapes) is 6.3 × 10^−4^ M cymoxanil. This is in excellent agreement with the performances of the NB-Gr/GC electrode, which may be used to check the initial concentration of the fungicide solution.

## 5. Conclusions

In this work, a graphene sample was chemically co-doped with nitrogen and boron heteroatoms and then employed for the electrochemical detection of a fungicide, cymoxanil. The dependence of cymoxanil peak current on the solution concentration (4 × 10^−5^–10^−3^ M) was measured under the optimized conditions (pH 6 BR). The limit of quantification (LOQ) was determined to be 4 × 10^−5^ M, while the limit of detection (LD) was 1.21 × 10^−5^ M. The modified electrode was also tested for successive measurements and it proved to have a good reproducibility. The amperometric technique was complementarily used to test the performances of the NB-Gr/GC electrode. The technique proved to be more sensitive, so both the limit of quantification and limit of detection were considerably smaller. Hence, LOQ was found to be 5 × 10^−6^ M, while LD was 1.51 × 10^−6^ M. In addition, the NB-Gr/GC electrode was also tested for its ability to detect CYM from a commercial product (CURZATE MANOX) by the standard addition method. According to data sheet specifications of CURZATE MANOX, the optimum concentration used for the treatment of various vegetables and fruits (potatoes, cucumbers, grapes) is 6.3 × 10^−4^ M cymoxanil. This is in excellent agreement with the performances of the NB-Gr/GC electrode, which may be used to check the initial concentration of the fungicide solution.

## Figures and Tables

**Figure 1 sensors-21-06630-f001:**
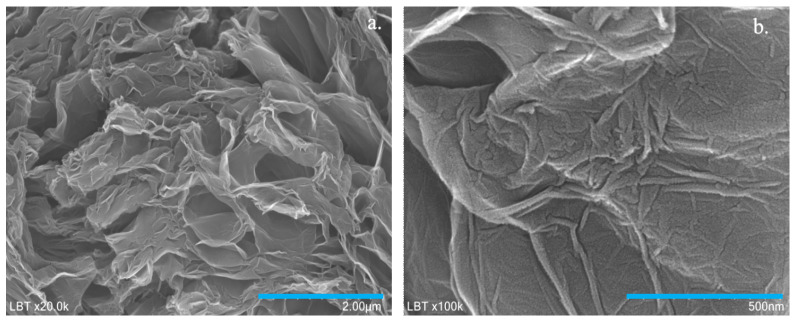
SEM micrographs with different magnifications of N,B-co-doped graphene sample; scale bar 2 μm (**a**); 500 nm (**b**).

**Figure 2 sensors-21-06630-f002:**
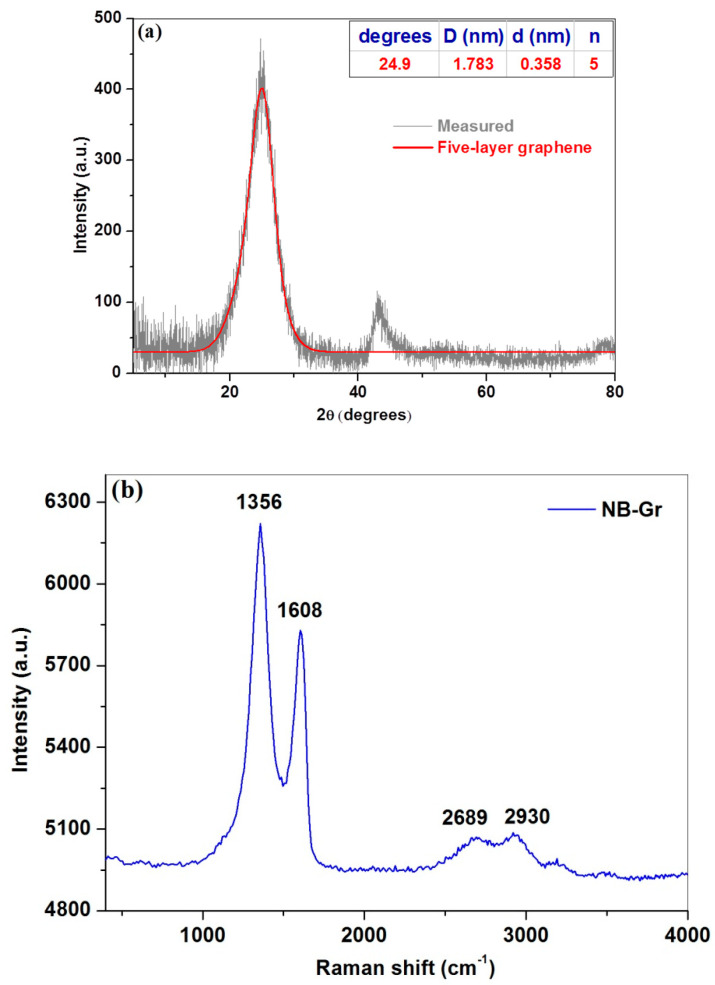
The XRD pattern (**a**) and the Raman spectrum (**b**) of N,B−co−doped graphene sample.

**Figure 3 sensors-21-06630-f003:**
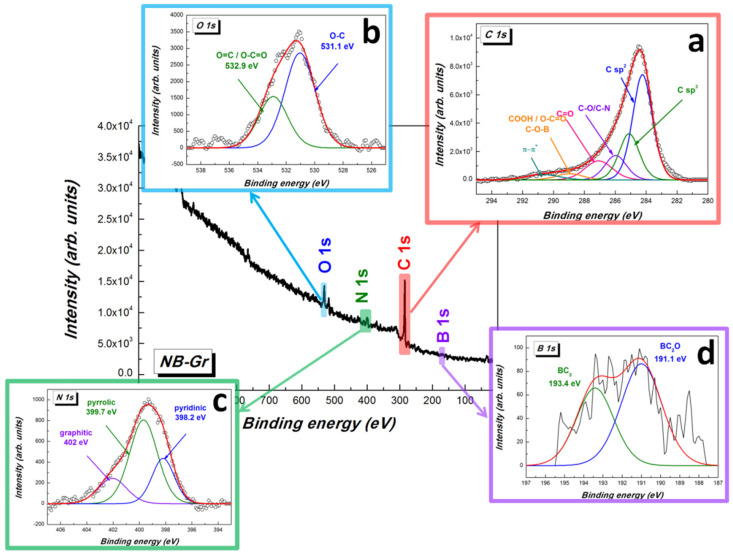
XPS survey spectrum and deconvolution of core level high-resolution C 1s (**a**) O 1s (**b**), N 1s (**c**) and B 1s (**d**) XPS spectra of N,B-co-doped graphene sample.

**Figure 4 sensors-21-06630-f004:**
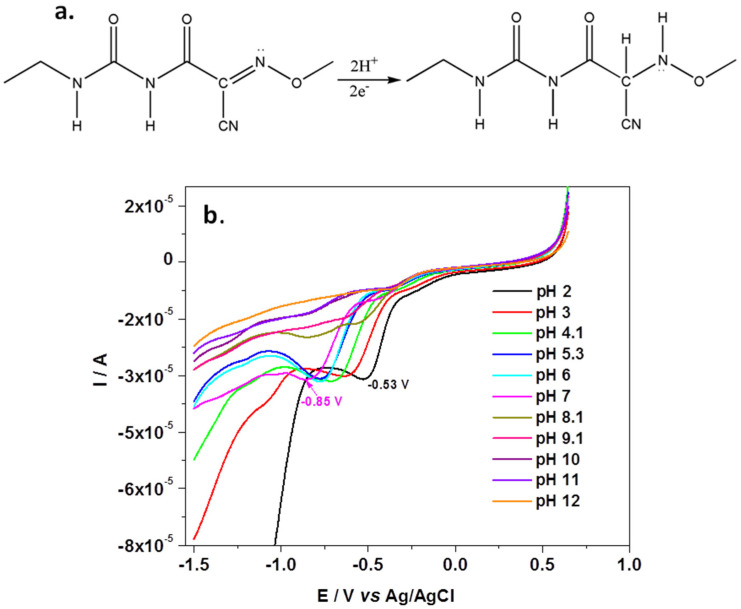
Electrochemical reduction of CYM (**a**); LSV recorded in BR solutions of various pH (2–12) each containing 10^−3^ M CYM; scan rate 10 mV/s (**b**).

**Figure 5 sensors-21-06630-f005:**
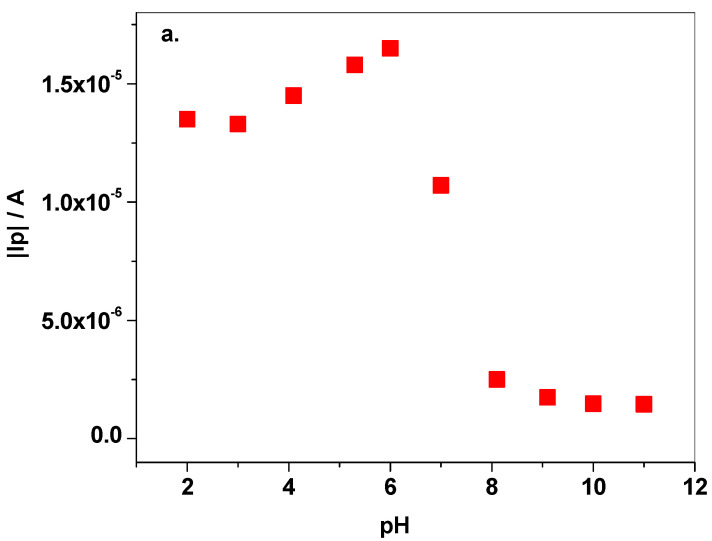
Variation of |I_p_| (**a**) and |E_p_| (**b**) with the pH of BR solution (pH 2–pH 12).

**Figure 6 sensors-21-06630-f006:**
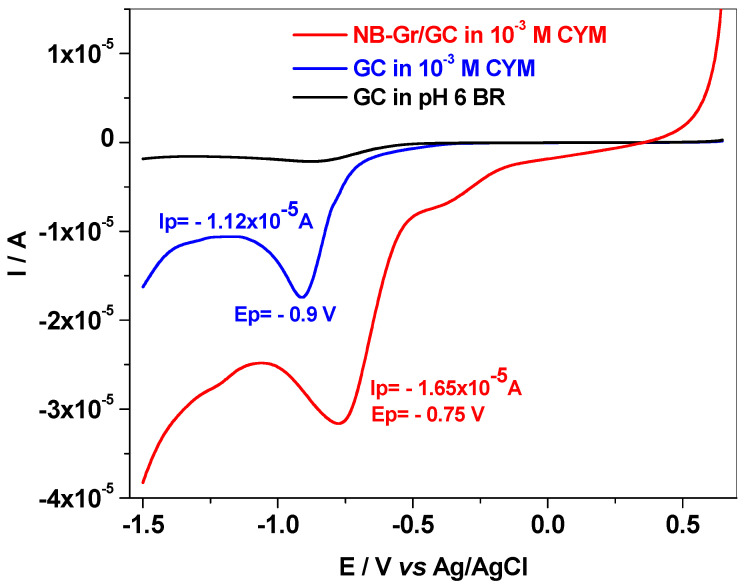
LSVs recorded with NB-Gr/GC electrode (red) and bare GC (blue) in pH 6 BR solution containing 10^−3^ M CYM; GC in pH 6 BR (black); scan rate 10 mV/s.

**Figure 7 sensors-21-06630-f007:**
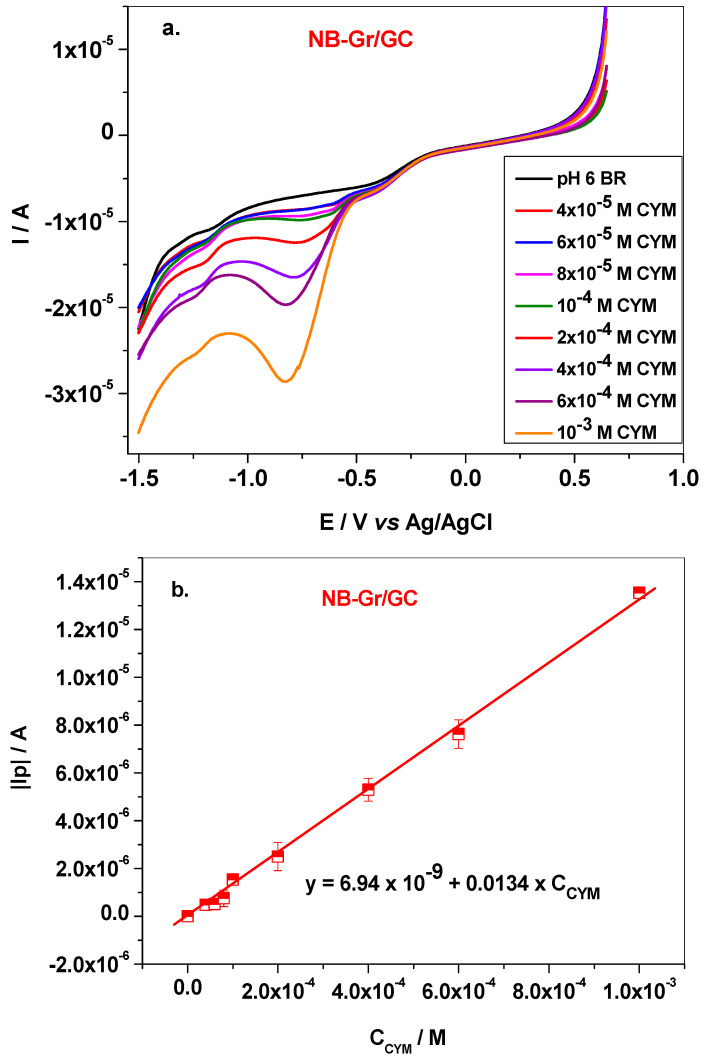
LSVs recorded with NB-Gr/GC electrode in pH 6 BR solutions, containing various concentrations (4 × 10^−5^–10^−3^ M) of CYM; scan rate 10 mV/s (**a**); the corresponding calibration plot within the linear range: 4 × 10^−5^–10^−3^ M (**b**).

**Figure 8 sensors-21-06630-f008:**
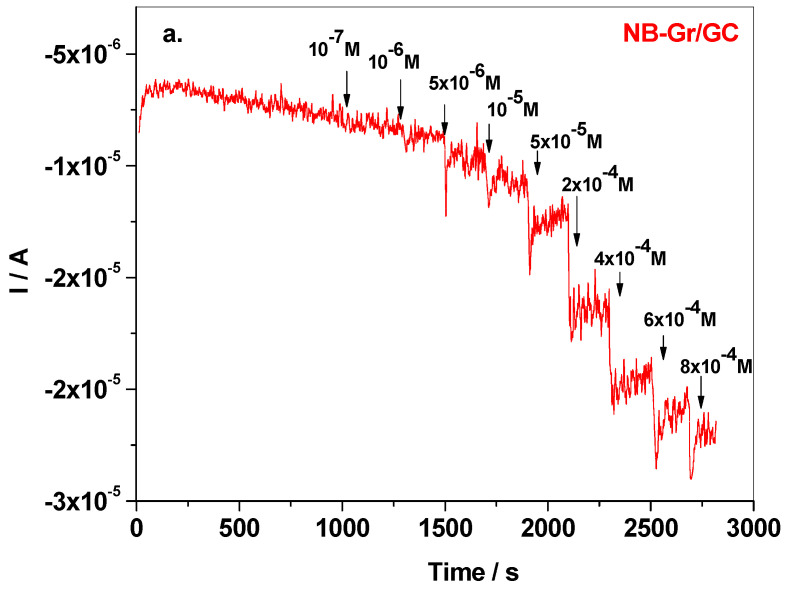
Amperometric curve recorded with NB-Gr/GC electrode in pH 6 BR supporting electrolyte, after the addition of known concentrations of CYM; applied potential −0.8 V vs. Ag/AgCl (**a**); the corresponding calibration plot within the linear range: 5 × 10^−6^–8 × 10^−4^ M (**b**).

**Figure 9 sensors-21-06630-f009:**
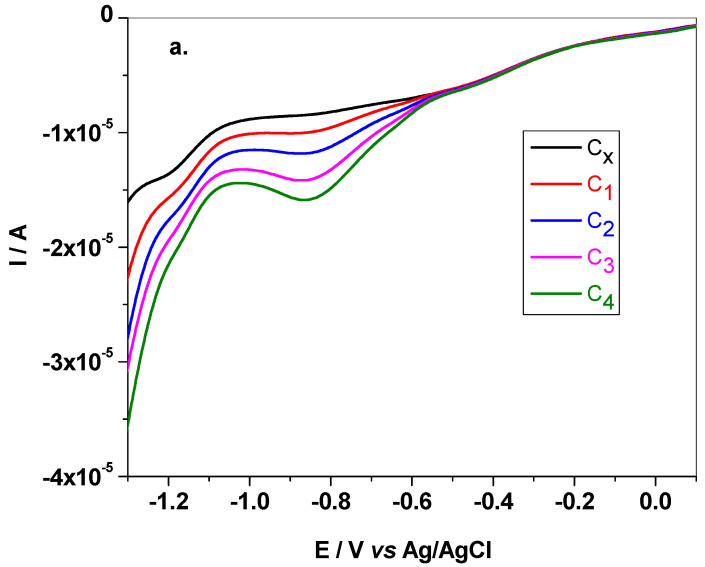
LSVs recorded with NB-Gr/GC electrode in CURZATE MANOX solutions containing various CYM concentrations (from 6.3 × 10^−5^ to 6 × 10^−4^ M); pH 6 BR supporting electrolyte; scan rate 10 mV/s (**a**); the corresponding calibration plot obtained from the standard addition method (**b**).

**Table 1 sensors-21-06630-t001:** The electrochemical performances of NB-Gr/GC electrode compared with those of other types of electrodes, found in the literature.

Electrode	Experimental Conditions	Method	Linear Range (M)	Detection Limit (M)	Ref.
HMDE	pH 7; BR	SWSV	1.2 × 10^−7^–9.85 × 10^−6^	3.58 × 10^−8^	[36]
CFRE	pH 4 BR:MeOH (9:1)	DPV	1 × 10^−5^–6 × 10^−7^	5.9 × 10^−7^ (quantification limit)	[43]
GCE	pH 7 BR: MeOH (9:1)	DPV	1 × 10^−5^–4 × 10^−7^	5.6 × 10^−7^ (quantification limit)	
NB-Gr/GC	pH 6 BR	LSV AMP	4 × 10^−5^–10^−3^ M 5 × 10^−6^–8 × 10^−4^ M	1.21 × 10^−5^ M 1.51 × 10^−6^ M	This work

HMDE—hanging mercury drop electrode; CFRE—composite fiber rod electrode; GCE—glassy carbon electrode; SWSV—square-wave stripping voltammetry; LSV—linear sweep voltammetry; AMP—amperometry; BR—Britton–Robinson buffer solution.

## Data Availability

Data will be provided upon reasonable request to the corresponding authors.

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
