# Peer review of "Hydrothermal Synthesis of Nitrogen, Boron Co-Doped Graphene with Enhanced Electro-Catalytic Activity for Cymoxanil Detection"

_sensors, 2021, doi:10.3390/s21196630_

Round 1

Reviewer 1 Report

The manuscript „Hydrothermal synthesis of nitrogen, boron co-doped graphene with enhanced electro-catalytic activity for cymoxanil detection” describes an experimental study to evaluate the potential and feasibility of glassy carbon electrodes modified with N,B-co-doped graphene for the detection of a commercially used fungizide (cymoxanil) by LSV.

There are a few concerns and remarks regarding the manuscript:

The paper concisely reports the experimental procedures applied, the results of the measurements, the data processing procedures. It draws traceable conclusions from the measurements and a verification experiment making use of a commercial formulation. There are very few shortcomings throughout the manuscript.

From line 178 (Legend of Fig.4), cymoxanil is eventually abbreviated by “CYM”, but this abbreviation was not introduced before. I suggest to define the abbreviation in line 66, where also the systematic chemical name of cymoxanil is given.

Line: 118-123: The XRD pattern (Fig. 2a) has been fitted using Gaussian profiles. Furthermore, a percentage of three- and six-layer graphene is given. The fitting of laboratory X-ray data with gaussians rather than Pearson-VII or Pseudo-Voigt profiles has to be regarded as an error and has to be corrected. Despite this is a severe experimental error, I regard this as a minor error in the context of the manuscript as no further conclusion is drawn from the result.

Furthermore, a reference for the technique applied to estimate the percentages of the differently layered graphenes is missing and should be added.

Line 220-242: The test experiment with a commercial product nicely describes the robustness and reliability of the presented technique. Nevertheless, it is not clear whether the robustness will stand in more field-like environments, as they are addressed in the Introduction (line 66-75). This should be mentioned briefly in the conclusions or an outlook section.

Reviewer 2 Report

Dear authors,

your manuscript is interesting, yet here are some comments 

-for the sample I would propose the expression: N,B-co-doped graphene

-enhance the "introduction" part, some theory on doping and on electrodes applications on chemical compounds (especially pesticides) would be helpful for the readers.

-never use a comma (,) before the "and" when for simple parathesis of similar things. Correct throughout the text

-provide some more information on instrumental techniques and methods applied (models, auxiliary parts, conditions, parameters set)

-its is mL, correct where needed. It is 180 °C, correct where needed.

-on SEM images I would prefere a continued line for scale instead of dots, it is safer for reading

-Fig. 3: is it safe to make comments for N1s peak? 

--Fig. 5: Ip dots shape an "S"-type curve or it's preferable not to say? The linear eqaution for Ep refers in the region pH=2-7 only I guess. Comment

-CYM stand for what? Avoid unnecessary/unfamiliar abbreviations because is tiring for the readers.

 You may also enrich the conclusion part of the manuscript. 

Reviewer 3 Report

The authors described a method for cymoxanil determination by linear
sweep voltammetry (LSV) and amperometry using a glassy carbon electrode modified with nitrogen, boron co-doped graphene, synthesized by hydrothermal method. This work could be of interest to the readers of Sensors. I recommend its publication after the following revision.

  1. For XPS, what is the peak between 700-800 eV?
  2. For figure 4, please explain the current which is larger than 0. What does this current come from?
  3. Are the sensors stable enough for reproducing the data?
  4. could the authors compare the present sensor with the literatures?
